# A Pilot Evaluation of mHealth App Accessibility for Three Top-Rated Weight Management Apps by People with Disabilities

**DOI:** 10.3390/ijerph18073669

**Published:** 2021-04-01

**Authors:** Erin Radcliffe, Ben Lippincott, Raeda Anderson, Mike Jones

**Affiliations:** 1Shepherd Center, Atlanta, GA 30309, USA; ben.lippincott@shepherd.org (B.L.); raeda.anderson@shepherd.org (R.A.); mike.jones@shepherd.org (M.J.); 2Department of Sociology, Georgia State University, Atlanta, GA 30302, USA

**Keywords:** mHealth, mobile applications, accessibility, usability, evaluation methods, user needs discovery, inclusive design

## Abstract

Growing evidence demonstrates that people with disabilities face more challenges in accessing healthcare and wellness resources, compared to non-disabled populations. As mobile applications focused on health and wellness (mHealth apps) become prevalent, it is important that people with disabilities can access and use mHealth apps. At present, there is no source of unified information about the accessibility and usability of mHealth apps for people with disabilities. We set out to create such a source, establishing a systematic approach for evaluating app accessibility. Our goal was to develop a simple, replicable app evaluation process to generate useful information for people with disabilities (to aid suitable app selection) and app developers (to improve app accessibility and usability). We collected data using two existing assessment instruments to test three top-rated weight management apps with nine users representing three disability groups: vision, dexterity, and cognitive impairment. Participants with visual impairments reported the lowest accessibility ratings, most challenges, and least tolerance for issues. Participants with dexterity impairments experienced significant accessibility-related difficulties. Participants with cognitive impairments experienced mild difficulties and higher tolerances for issues. Our pilot protocol will be applied to test mHealth apps and populate a “curation” website to assist consumers in selecting mHealth apps.

## 1. Introduction

Over 80% of people with disabilities in the US live with chronic conditions that compound the effects of disability on health and function. There is growing evidence that people with various types of disabilities face more challenges in accessing healthcare, as well as preventive health and wellness management resources [1,2,3]. Mobile health (mHealth) is emerging as an important tool for managing chronic health conditions, but people with disabilities have not been a primary target for mHealth app development, and accessibility needs are often overlooked or disregarded [4]. Common accessibility needs for people with dexterity or visual impairments include app compatibility with alternative access software and assistive devices [5,6]. People with visual and cognitive impairments often need to use magnification software, custom contrast, color adjustments, and auditory alerts to access their mobile devices and apps [7,8]. Rapid proliferation of mHealth apps could further increase health disparities between people with and without disabilities if apps disproportionately benefit non-disabled populations. Early evidence suggests that disparities in adopting mHealth apps among people with disabilities may already exist [9,10]. While many popular, commercially available “diet and exercise” apps have been evaluated for efficacy and general ease of use, there is limited information available about accessibility and usability by people with disabilities [11,12,13,14,15]. Given this scarcity of information, we set out to develop a model for evaluating mHealth apps for accessibility. We conducted a concurrent mixed methods usability and accessibility evaluation of three top-rated, commercially available mHealth apps that assist users in reaching diet, exercise/fitness, and weight management goals.

Our objectives for this study were to develop and test a prototype app accessibility testing protocol and to disseminate findings via an app curation website that provides information to people with disabilities about mHealth app suitability, based on their needs, impairments, and mobile device access methods. Our findings are also intended to inform recommendations to mobile app developers for how to resolve common accessibility issues and fulfill inclusive design requirements.

## 2. Materials and Methods

### 2.1. mHealth App Selection Process

We identified the three top-rated, commercially available weight management apps, based on subject-matter-expert recommendations [15,16] and comprehensive reviews evaluating popular mHealth apps for general usability and effectiveness [13,14,17,18]. The three apps were: MyFitnessPal, Lose It!, and FatSecret [19,20,21]. We chose these three apps based on the following selection criteria:High consumer review ratings evenly reported on both the iOS App Store and in Google PlayHigh download/installation user-base on both iOS and Android OS devicesSuggested by subject matter experts in content evaluation methodology for commercial mobile health apps supporting healthy eating and physical activity [15,16]Identified by published research reviewing smartphone apps aimed towards promoting healthy nutrition [13], physical activity [14,17], and weight management [18]Commercially available on both iOS and Android OS devicesFree with in-app purchases for both iOS and Android device users

Ten mobile apps met the minimum thresholds for consideration: MyFitnessPal, Lose It!, Noom [22], FatSecret, WW (Weight Watchers) [23], MyPlate [24], VA MOVE! Coach [25], MyNetDiary [26], My Diet Diary [27], and SparkPeople [28]. We chose the following three apps, as they best satisfied our selection criteria:

MyFitnessPal is a weight management app that focuses on personalized diet and exercise tracking [19]. Two efficacy trials found that MyFitnessPal, compared with standard interventions for dietary self-monitoring and weight management, was equivalent in effectiveness and offered reliable compatibility with commercial hardware (e.g., Fitbit and Garmin activity trackers) to support participant diet and fitness goals [29,30].

Lose It! is a weight-loss app that supports goal-setting and nutrient tracking [20]. In a study evaluating data captured in Lose It!, results indicate successful weight loss outcomes in distinct subgroups and identify characteristics among subgroups who experienced more success in reaching their goals [31].

FatSecret is a calorie-counter app that enables food and exercise logging paired with a peer support network to reinforce users’ weight-loss goals [21]. A comparison study evaluating standard versus mHealth app supported self-monitoring demonstrates preliminary evidence suggesting that diet and fitness tracking using FatSecret may produce more successful weight management outcomes [32].

### 2.2. App Accessibility Testing Protocol

Our app accessibility testing protocol was informed by a review of prior research regarding mHealth app evaluation methods, rapid user-testing methods, and a standardized system for quantifying usability feedback [33,34,35]. We evaluated accessibility and usability issues by asking participants to complete the following tasks within each of the three apps:Explore: initial familiarization to app screens, functions, and navigational elementsCore Tasks: logging meals, physical activity, and diet or fitness goals

This approach allowed us to assess general ease of use, application compatibility with built-in and external accessibility aids, and the impact of accessibility issues on each user’s ability to perform primary tasks within each app. We used standardized evaluation methods from two existing measures of software usability and accessibility—the System Usability Scale (SUS) [35] and Tierney’s 7-min Accessibility Assessment and App Rating System [34].

#### 2.2.1. System Usability Scale (SUS)

The SUS is a standardized measure providing a quantifiable assessment of subjective usability assessments [36,37]. The SUS can provide reliable results with small sample sizes [38,39]. In multi-survey studies, the total SUS score, with a maximum of 100, averages at 69.69 with a range of 30.00–93.93 [38].

After completing each task in our app assessment, participants were asked to score the following 10 items on a 5-point Likert scale ranging from “strongly disagree” (1) to “strongly agree” (5):I think that I would like to use this system frequently.I found the system unnecessarily complex.I thought the system was easy to use.I think I would need the support of a technical person to be able to use this system.I found the various functions in this system were well integrated.I thought there was too much inconsistency in this system.I would imagine that most people would learn to use this system very quickly.I found the system very cumbersome to use.I felt very confident using the system.I needed to learn a lot of things before I could get going with this system.

#### 2.2.2. Tierney’s 7-Minute Accessibility Assessment and App Rating System

Through interview-administered surveys with both closed and open-ended questions, we asked participants to describe their difficulties while using the app and share their ideas for changes to improve app accessibility and usability. Participants responded to the following three items, adapted from Tierney’s app rating system [34]:Please rate your experience completing [name of task or task section], based on ease of use, on a scale of 1 to 5 stars using the following rating key:
○1 Star: Very Difficult (lowest tolerance for issues)—Unable to complete most tasks independently; Prohibitive confusion or frustration○2 Stars: Difficult—Unable to complete a few tasks independently; Significant confusion or frustration○3 Stars: Moderate Ease/Difficulty—Able to complete all tasks independently; Moderate confusion or frustration○4 Stars: Easy—Able to complete all tasks independently; Infrequent confusion or frustration○5 Stars: Very Easy (highest tolerance for issues)—Able to complete all tasks independently; Minimal to no confusion or frustrationWhile completing [name of task or task section], was there anything of note, that made the process difficult? If yes, please describe the source(s) of difficulty.While completing [name of task or task section], was there anything of note, that you would want to change about the process or app interface to make accomplishing the tasks easier? If yes, please describe what you would like to see changed.

We made minor adaptations to the wording describing Tierney’s app rating system to improve the simplicity, clarity, and standardized readability of each item for participants and people unfamiliar with principles of mobile device accessibility.

The 5-star accessibility ratings provide valuable context, when paired with the qualitative feedback gained from targeted questions asked following each task completion. The 5-star ratings also serve as an analog to commercial product-review consumer ratings (typically seen in app stores and on sites like Amazon.com).

### 2.3. Participant Recruitment and Testing Procedure

To evaluate the utility of our concurrent mixed methods assessment protocol in generating useful app accessibility and usability information, we recruited a diverse sample of people with disabilities: a third who have dexterity impairments, a third who have visual impairments, and a third who have cognitive impairments. We recruited participants via an emailed recruitment letter, which outlined the study’s objectives and participation requirements, and provided basic information about the apps being assessed. Interested respondents were sent a link to an online informed consent form and intake survey that collected information about their demographics and assistive technology usage to access their mobile device.

Before beginning the assessment, participants downloaded the app from the Apple or Google Play store onto their personal mobile device. Participants were then asked to complete a guided series of accessibility testing tasks, adapted from the Tierney assessment model [34]. We asked participants to describe their experience completing the tasks within each app, using the SUS and Tierney assessment measures. We recorded participants’ responses in a secure data collection and management tool (REDCap).

To supplement each participant’s accessibility and usability assessment, one of the researchers observed the participant’s interactions with the app and completed a “significance and impact” assessment, to document unstated but notable accessibility/usability issues experienced during task completion. We combined the observational assessment with the participant’s qualitative feedback, overall Tierney “5-star” accessibility rating, and total SUS scores for each app.

## 3. Results

### 3.1. Participant Characteristics and Assistive Technology Usage

Table 1 summarizes the nine participants’ reported demographic characteristics, collected in the study’s initial intake survey. There are equal number of participants across impairment categories with an average age of 41 years old and an average time with impairments at 16 years. There are nearly equal numbers of males and females, and the average annual income is USD 22,000 with a notable range (less than USD 15,000 to between USD 100,000 and 150,000). This annual income distribution did not notably vary by impairment group. Slightly more than half of respondents use an iOS mobile platform, and the rest use Android mobile operating systems.

Figure 1 summarizes the nine participants’ reported usage of assistive technology, accessibility software, and built-in general consumer aids while using mobile devices, also collected in the study’s intake survey. Each participant relied upon at least one form of technical accommodation to use their mobile phone. The represented technology usage distribution is color-sorted by participant impairment category. Of the eleven assistive technology and general consumer aid options, the most commonly used technologies were Intelligent Personal Assistant (66.6% of the dexterity group; 100.0% of the visual group; 33.3% of the cognitive group) followed by Word Prediction and Autocorrection Software (66.6% of the visual group; 66.6% of the cognitive group), and Auditory Alerts (66.6% of the visual group; 33.3% of the cognitive group). The least used assistive technology is Braille Application (33.3% of the visual group), followed by Accessible Keyboard (66.6% of the visual group), and Custom Contrast or Color Adjustment (66.6% of the cognitive group). Of note, participants with dexterity impairments used only two of the eleven technology aids: Assistive Devices/Alternative Access Software (66.6% of the dexterity group) and Intelligent Personal Assistant (66.6% of the dexterity group), while participants with visual impairments used nine and participants with cognitive impairments used seven of the eleven listed technology aids.

### 3.2. Quantitative Results

#### 3.2.1. Quantitative SUS Score Results

Table 2 summarizes the total SUS scores per disability group for all app testing sessions. One-way analysis of variance (ANOVA) was used to compare mean differences between disability groups on SUS scores (0–100) for the task-completion section of each app. Analysis of variance is a common statistical test which examines the mean distributions between three or more unrelated groups to determine if the groups’ averages are significantly different from each other. When an ANOVA test results in an alpha below 0.05, there are significant differences between the averages of the groups. The F statistic is the outcome of the ANOVA paired with the degrees of freedom (df) to determine statistical significance, which is commonly referred to as the *p* value. *p* values less than 0.05 are statistically significant (Sig.). Additionally, the number of respondents in each group is reported (N) [40].

Table 2 illustrates that participants with visual impairments had greater difficulty in completing evaluation tasks for all three apps, compared with other disability groups. These between-group differences were statistically significant for the “Explore” activities with the Lost It! and FatSecret apps. For MyFitnessPal, there were no notable differences in SUS total scores between disability groups while exploring and completing core tasks in the app. For Lose It!, participants with visual impairments reported significantly lower total SUS scores than participants without visual impairments while initially exploring the Lose It! mobile interface (F(9) = 14.286, *p* = 0.005). Similarly, participants with visual impairments reported significantly lower scores than participants with dexterity or cognitive impairments (F(9) = 17.028, *p* = 0.003) while initially exploring the FatSecret app. The overall pattern is clear and consistent across app evaluations: where substantive differences between impairment groups exist, participants with visual impairments report the lowest overall usability scores and the most significant accessibility issues, compared to participants without visual impairments, especially while initially exploring an app interface.

#### 3.2.2. Quantitative Tierney Rating Results

Figure 2 presents mean Tierney 5-Star accessibility ratings, representing participants’ perceived ease of use for each mHealth app, overall, and for each participant disability group. On average, participants with dexterity impairments reported higher Tierney Accessibility ratings, indicating they experienced the least amount of difficulty (as a group) while using each of the three apps (mean ranges from 4.4 to 4.9 out of 5 across apps). Participants with visual impairments reported the lowest Tierney accessibility ratings, indicating they experienced the most difficulty using the three apps (mean ranges from 2.8 to 3.3 out of 5). Participants with cognitive impairments consistently provided medium-high Tierney accessibility ratings (mean ranges from 4.1 to 4.4 out of 5) for each mHealth app.

On average across impairment groups, FatSecret was rated highest for accessibility (4.1), followed by Lose It! (4.0), while MyFitnessPal was rated lowest (3.8) for accessibility. Participants with dexterity impairments rated the Lose It! app highest for accessibility (4.9). Participants with visual impairments and cognitive impairments rated FatSecret highest for accessibility on Tierney’s app accessibility rating scale (3.3 and 4.4, respectively).

There are no published studies employing Tierney App Accessibility Ratings. However, Tierney notes the level of associated accessibility with each rating as follows [34]:1 Star: Very Low—Severe accessibility issues and overall inattention to accessibility will prevent some users from using the app.2 Stars: Low—Accessibility issues will prevent some users from completing core tasks in the app, due to poor overall accessibility quality.3 Stars: Moderate—Accessibility issues may prevent some users from completing non-core tasks. Users may experience significant annoyances and confusion, due to lack of attention to detail across the application.4 Stars: High—Some minor impact issues may exist, but no accessibility issues prevent user from completing core tasks, despite annoyances.5 Stars: Very High—Low incidence of issues. Some minor impact issues may exist, but no accessibility issues prevent user from completing core tasks.

### 3.3. Qualitative Results

While completing the “Explore” and “Core Tasks” sections of our app accessibility testing protocol, assessing MyFitnessPal, Lose It!, and FatSecret, the following themes emerged:

Participants with dexterity impairments experienced significant difficulties, especially in instances where functional elements were either not available or more cumbersome for them to use. Difficulties occurred when built-in voice controlled mobile accessibility features, utilizing vocally prompted commands or numerical grid overlays, could not successfully manipulate standard app elements, or when clickable app elements were unresponsive to a user’s assistive device inputs.

One participant with dexterity impairments using iOS Voice Control (a built-in accessibility feature that enables Apple phone use through voice commands) was unable to easily navigate within MyFitnessPal because the accessibility feature’s overlay could not consistently access clickable buttons and fields within the app interface [41]. Another participant with dexterity impairments, using a Bluetooth-paired GlassOuse assistive device (a wearable gyroscope-based system worn like glasses to enable cursor control using head movements and clicking action using a connected bite-switch), was unable to control the MyFitnessPal app interface, because many of the entry fields were unresponsive or overly sensitive to the device inputs [42].

Participants with visual impairments experienced the most usability and accessibility difficulties, especially while exploring each app, due to frequently encountered instances of unlabeled buttons, icons, navigational elements, screen headings, and entry-field labels within each app. These participants, using a screen-reader to access their mobile devices, often got stuck, lost, discouraged, and confused while using each app, and they often required assistance to gain sufficient context to locate and use standard functions during and between tasks.

Alt Text was inconsistently integrated throughout each app, and participants with visual impairments using screen-readers were unable to read or access wrapping menus, graphic data, and most media on MyFitnessPal’s home screen (e.g., blog posts, workout videos, and recipes). Graphical representations of data in Lose It! were also not accessible to participants with visual impairments using screen-readers.

Participants with cognitive impairments experienced moderate difficulties or confusion while completing sub-portions of each task completion section per app, but they tended to have higher tolerances for each experienced issue and were consistently able to complete each task in the protocol without assistance. These participants struggled to consistently intuit the purpose of certain buttons with unclear text-labeling and sometimes struggled to read elements with low-contrast text-to-background labels and icons in each app. They frequently requested guidance, commenting that an info tab or hyperlink to instructions for how to use important app features would be helpful to them.

## 4. Discussion

As mHealth app usage and development continues to grow, it is increasingly important that apps be accessible and usable for the widest range of human abilities. Harrison et al. conducted a review of mobile usability models and notes that usability is often measured by three core attributes: effectiveness, efficiency and satisfaction, though other important attributes, such as cognitive load and utility tend to be disregarded, despite their likely influence on the level of a mobile application’s usability [43]. This study introduced a more comprehensive usability model designed to address the limitations of existing models when applied to mobile devices [43]. This multifaceted usability framework can be applied in parallel with publicly available accessibility guidelines to understand the range of user needs that must be considered when developing, assessing, and selecting suitable mHealth applications.

Various sets of accessibility guidelines have been established, such as the World Wide Web Consortium (W3C) Web Content Accessibility Guidelines (WCAG) 2.0 and the User Agent Accessibility Guidelines (UAAG) 2.0 published by the W3C Web Accessibility Initiative (WAI) [44,45,46]. Notably, the two leading multinational technology companies that design and develop the majority of the world’s mobile apps, Google LLC and Apple Inc, have published extensive amounts of information for developers on how to develop, improve, and test for accessibility on Android and Apple mobile applications [47,48,49,50,51,52]. Both Google and Apple have also created and enabled the integration of compatible mobile accessibility features that are now either built-in or publicly available for all Android and iOS mobile device users [53,54]. These native consumer software accessibility features aim to support mobile device users with visual, mobility/dexterity, cognitive, and hearing impairments. However, as noted in our results and in the results of other larger-scale studies evaluating the accessibility and usability of mobile apps, accessibility issues remain prevalent in commercial mobile apps, indicating the need for informed support for both app consumers with disabilities and mobile app developers [55,56,57].

A study completed by Yan et al. assessed the level of accessibility in 479 Android mobile apps using an automated accessibility evaluation tool and found a pervasive incidence of accessibility issues across the sample of evaluated mobile apps, primarily caused by “missing element descriptions, low text color contrast, lacking element focus,” insufficient spacing between elements, and “less than minimum sizes of text and elements” [55]. Eler et al. also explored automated test generation and employed a mobile accessibility testing tool to evaluate 73 apps for accessibility issues affecting users with visual impairment [56]. This study reveals that both manual accessibility testing methods and the automated accessibility checker detected numerous but differing accessibility issues in the sample of evaluated mobile apps, even when mobile accessibility aids were enabled during testing [56]. These results highlight the value of both automated and manual mobile app accessibility evaluations as complimentary methods.

Results obtained through automated accessibility evaluation tools align with insight gleaned from our small-scale, in-person app accessibility testing results, as both discovered shared themes regarding the frequency and nature of common mobile app accessibility issues. Given that our protocol detected accessibility issues experienced by people with disabilities in real time, we were able to not only detect accessibility non-conformance, but also observe and note the impact that the accessibility issues made on the app user’s ability to complete tasks for intended use cases within the apps we evaluated.

Aguado-Delgado et al. presented a comprehensive approach for evaluating the accessibility and usability of mobile apps informed by W3C recommendations and usability heuristic evaluation methods [57]. Similar to our protocol design, app accessibility evaluations in this study involved observing participants’ completion of guided tasks within the mobile application’s relevant scenarios in real time while documenting observed and reported accessibility issues along with each issue’s severity and impact on the participant’s successful completion of tasks within the app [57]. The results of this work conclude that a comprehensive evaluation of mobile app accessibility and usability, using a similar systematic procedure informed by prior art, can be completed by non-expert evaluators “in a short period of time” without the use of automated accessibility evaluation tools [57].

Our pilot protocol enables an informed and practical approach for assessing the accessibility and usability of mHealth apps with participants with disabilities to generate actionable insight for mHealth app consumers with similar impairments to assist their selection of optimal mHealth apps. While our small-scale quantitative analyses provide a basis for comparison of accessibility and usability for each evaluated app by disability group, our participants’ qualitative feedback provides valuable information to help consumers select the mHealth app that may be best suited to their needs and abilities. This systematically collected qualitative data also identifies specific areas within each evaluated app that require attention and effort from app developers towards accessibility and usability improvements.

If any aspect or function within a weight management app (e.g., login, navigational elements, meal diary entry, exercise logging, graphical data) is not accessible, a user is likely to become frustrated or discouraged, increasing their likelihood of abandoning the app and not benefitting from the support it was designed to offer. Consequently, lacking or inconsistent accessibility throughout an mHealth application risks reinforcing the preexisting disparity between people with disabilities and non-disabled populations in accessing publicly available health and wellness resources. Our methodically acquired qualitative results inform guidance to app developers on how to specifically improve each app’s accessibility and usability for participants with dexterity, visual, and cognitive impairments, as well as for general app consumers.

### 4.1. Opportunities for Improved Mobile App Accessibility:

To address the needs and use-cases of people with dexterity impairments who use built-in Voice Control/Voice Access accessibility features to control their mobile devices, we recommend that app developers design and test app production interfaces with accessibility feature overlay constraints in mind, to ensure that key functional buttons and navigational elements are accessible to app users using built-in accessibility features [46,47,49,51]. We also suggest that app developers enable the options to manually enter values via voice entry or a standard on-screen keyboard, rather than requiring certain data-entries to be submitted solely via a scroll-wheel. For iOS Voice Control users, entry fields requiring typing could be simplified by enabling users to enter data via spoken entries. To address the needs of people with dexterity impairments who use switch-control devices to operate their mobile devices, we recommend that developers iteratively test and refine functional element and field sensitivities within the app to translate assistive device enabled user-inputs more accurately [46,49,51].

To address the needs of people with visual impairments using screen-readers, we recommend that developers incorporate consistently integrated Alt Text for all functions, navigational elements, graphical representations, and data-entry fields [44,46,48,50,52]. We specifically recommend that app developers provide clearly labeled, and consistently located “back” and “next” buttons within each entry field and sub-app screen to provide visually impaired users with sufficient context for successful app navigation and use. To make app use more inclusive for participants with visual impairments, developers will also need to incorporate sufficient Alt Text in visual data representations (e.g., graph and chart elements) to enable app users utilizing a screen-reader to read, interact with, and gain equal benefits from graphical app features [46,50,52].

To improve ease of use and reduce preventable confusion or difficulty for app users with cognitive impairments, along with general consumers, we recommend that developers focus on incorporating clear and consistent labels for buttons, icons, and fields along with high text-to-background contrast when text is used to explain an element’s function [43,46,47,50]. Based on feedback from participants with cognitive impairments, we also suggest that app designers enable the capacity for users to show/hide login credential entries while typing, so that users can temporarily note what they have already typed during the data-entry process, rather than having to restart an entry each time they lose track of their entry field completion progress. Providing these options would likely benefit general consumers in increasing app ease of use [43,47,50].

Inconsistency in mobile interface functions, labels, displays, and entry-field types in an app can cause significant confusion and uncertainty among all app users, regardless of impairment type. To maximize usability during app use for consumers with various ability levels, developers must design apps as cohesive systems with unified structure, process-flows, and themes.

### 4.2. Limitations

Given the small participant sample size, typical of usability testing studies, results obtained through our quantitative analyses have low statistical power. Additionally, the small sample of participants is diverse both in demographic and impairment characteristics. This participant variance influences the consistency of both our quantitative and qualitative findings. Measured results and SUS score differences between impairment groups per mHealth app are not intended to infer quantitative trends, but more-so, are meant to provide objectively measured context when paired with richer, subjective qualitative analyses of participant experiences.

### 4.3. Future Work

As noted in the introduction, the objective of this study was to develop and test a protocol that can be used to systematically gather input from people with disabilities on app accessibility and usability, in order share ratings and insights with other prospective users with disabilities. To this end, we are building an app curation website, www.theappfactory.org, where the app testing results will be published. The website will have three purposes:To provide app users with disabilities clear ratings and information on accessibility issues they may encounter while using each evaluated app, based on their disability or the assistive technology they may use to access their mobile devices. This information is meant to help app users make informed decisions when selecting an mHealth app to meet their needs.To invite and enable app users with disabilities to provide their own accessibility ratings and “consumer reviews” for selected mHealth apps.To make recommendations to developers on steps they can take to improve their app’s accessibility and usability for people with disabilities.

Based on a review of the current state, opportunities and challenges, and future needs of mobile health for people with disabilities, we plan to refine and apply our app accessibility and usability testing methodology towards the evaluation of a diverse set of mHealth/mRehab apps focused on supporting various aspects of health and wellness [4,58,59]. Informed by results from a recent survey of user needs and preferences for mHealth apps by people with disabilities (physical, cognitive, sensory, emotional/psychological, and speech), our future work will focus on testing the accessibility and usability of the following mHealth/mRehab app categories [60].

Apps that track and support individualized goals and health habits;Apps that track multiple areas of health in one app;Clinical portal apps that enable personal health information management, appointment scheduling, access to lab results and vital sign tracking, and correspondence with healthcare providers (e.g., MyChart);Apps that support stress, PTSD, and mental health management;Apps that track and support healthy sleep patterns;Apps that support diabetes management via blood pressure, blood sugar, and heart rate monitoring;Sensor-enhanced activity monitoring apps that support remote physical rehabilitation;Exercise apps for wheelchair users.

These app categories are prioritized based on the reported mHealth apps most used or deemed valuable and necessary by survey respondents with disabilities [60]. Though we initially reasoned that apps supporting chronic health condition management would be important to include, survey resulted indicated that symptom and disease management apps were the least commonly used by respondents with disabilities [60]. In the next phase of this project, we will focus on the first two bullets listed above and aim to evaluate apps that track multiple areas of health to support individualized goals and health habits. We are looking forward to testing the accessibility and usability of this next identified collection of popular mHealth apps by people with disabilities.

## 5. Conclusions

Study results inform guidance for consumers with disabilities in choosing mHealth apps that meet their needs, and findings reveal common accessibility issues to inform app developers of key inclusive design requirements. Accessibility issues in popular mHealth apps for weight management hinder people with disabilities from successfully using and fully benefiting from these apps’ support in reaching diet, weight, and exercise goals. Maintaining a diet and exercise regimen for people with disabilities is more important now than ever during the COVID-19 pandemic, as many people have been unable to get out and enjoy their normal workout routines or easily access healthy food options. Apps continue to be developed at a rapid pace, and currently, over 2.8 million apps are available on Google Play while over 1.9 million apps are available in the Apple App Store [61]. Fulfilling inclusive design requirements often enables more positive usability experiences for both impaired and general consumers, and app developers need technical instruction to ensure their mobile interfaces are designed for accessibility to enable use by the widest range of human abilities. Moving forward, our evaluation protocol will be used to assess additional mHealth apps and populate an app “curation” website to assist consumers in locating accessible and usable mHealth apps and to inform developers of opportunities for improved mobile app accessibility.

## Figures and Tables

**Figure 1 ijerph-18-03669-f001:**
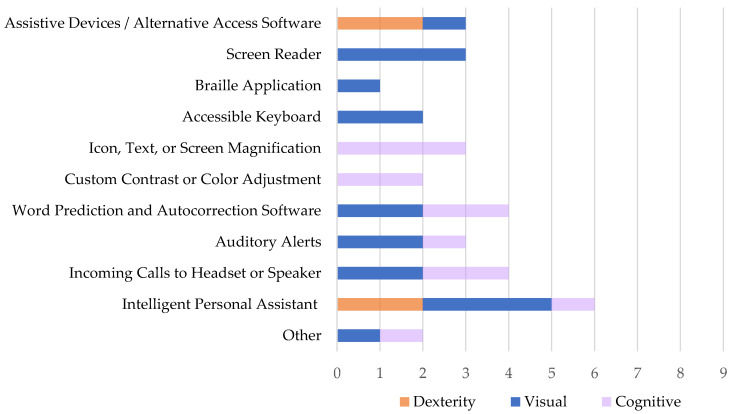
Assistive Technology Usage by Impairment Category (N = 9).

**Figure 2 ijerph-18-03669-f002:**
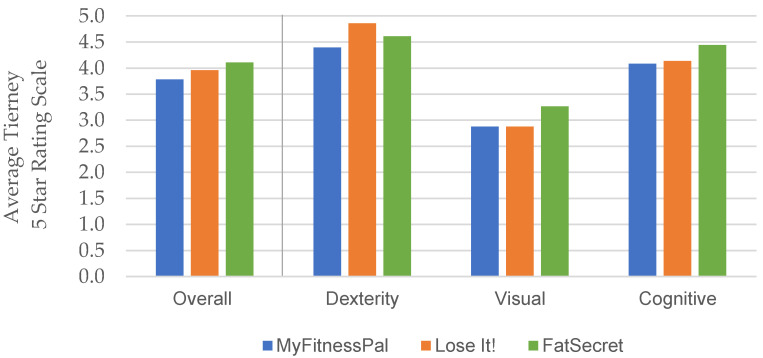
Mean participant Tierney ratings (1–5) both overall and by disability category per mHealth app.

**Table 1 ijerph-18-03669-t001:** Participant Demographic Characteristics (N = 9).

Demographic Variable	Participant Sample Characteristics
Impairment Category	3 Dexterity, 3 Visual, 3 Cognitive
Age (years)	Mean: 41; range: 25–64
Time with Impairments (years)	Mean: 16; range: 1–37
Gender	4 Female, 5 Male
Race/Ethnicity	1 Black, 8 White; 2 Hispanic/Latino
Level of Education (years)	Mean = 16; range = 14–18
Annual Income	Mean: USD 22 K; range: <USD 15 K to USD 100–150 K
Mobile Platform	5 iOS, 4 Android

**Table 2 ijerph-18-03669-t002:** SUS Results—Mean participant SUS Scores (0–100) across three apps, one-way analysis of variance (ANOVA).

mHealth App	Task Completion Section	Dexterity (N = 3) ^1^	Visual (N = 3)	Cognitive (N = 3)	ANOVA Results		
		M	M	M	F	P	Sig.
MyFitnessPal (N = 8, df = 7)	Explore	77.5	44.2	70.0	0.853	0.480	ns
	Core Tasks	70.0	61.7	78.3	0.812	0.495	ns
Lose It! (N = 9, df = 8)	Explore	83.3	31.7	81.7	14.286	0.005	*
	Core Tasks	89.2	47.5	82.5	3.601	0.094	ns
FatSecret (N = 9, df = 8)	Explore	94.2	41.7	89.2	17.028	0.003	*
	Core Tasks	90.8	57.5	77.5	2.593	0.154	ns

^1^ Dexterity N = 2 for MyFitnessPal and N = 3 for Lose It! and FatSecret. N = disability group size, df = degrees of freedom, M = Mean, F = F statistic, P = *p* value, Sig. = statistical significance level, ns = not significant; * *p* < 0.05.

## Data Availability

The data presented in this study are available upon request from the corresponding author at erin.radcliffe@shepherd.org. The data are not publicly available due to data privacy ethics.

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
