# Peer review of "A Pilot Evaluation of mHealth App Accessibility for Three Top-Rated Weight Management Apps by People with Disabilities"

_ijerph, 2021, doi:10.3390/ijerph18073669_

Round 1

Reviewer 1 Report

The paper proposes a model for evaluating mHealth apps for accessibility, which was tested using three apps (MyFitnessPal, Lose It!, and FatSecret) available for both iOS and Android platforms. 

The proposed approach is interesting and the topic is relevant for the community. However, the paper has some major drawbacks:
- Abstract:
please avoid to cite references  
- 1. Introduction :
Provide the general framework and the relevant references. I don't have comments on this section. 

- 2. Materials and Methods
Why have you chosen these three apps? Which are the criteria you've used? A proper discussion should be included. It can be of some use to list all the 23 examined apps (maybe in a supplementary material?)

lines 73-75: "Our app accessibility testing protocol was informed by a review of prior research 73 regarding mHealth app efficacy and usability evaluation methods, rapid user-testing 74 methods, and a standardized system for quantifying usability feedback [21-28]" 
Too much information is collapsed into a single sentence; therefore, it becomes hard to understand. Please rephrase it, expanding the different parts. FInally, avoid massive citations "[21-28]"; use proper citation where needed. 

lines 77-79: do a proper list, that's your protocol, it deserve a proper subsection and explanation.

Section 2 should be reorganized and divided into subsection to better encapsulate information (new lines are not sufficient to this purpose). I suggest you to use subsection for each metric, where you describe them in detail; a "Protocol" subsection where you summarize the proposed protocol (which is the innovative part you propose in this paper).

lines 129-131: population description may be put together (in Sec. 3) with lines 154-161 and table 1.

- 3. Results:
lines 170-177 what is n? each symbol must be always explained. I would also expand this paragraph to enhance readability. 

Sec. 3.2 is too short, I suggest you to put lines 185-187 within sec. 3.1 and make Sec. 3.2.1 sec 3.2 App Accessibility Testing Results. 

Anova metric should be explained in detail 

Table 2: what is df? what means the column Labels (M,F,P, sig., N)? please explain in the text and in the captions

line 200: "while completing either task completion section". remove either or rephrase. What is Task completion section? both "explore" and "core tasks" or just one of them?

Fig. 2 is hard to understand and confusing. Never encapsulate a table into a figure. I suggest you to deeply rethink Fig. 2 to enhance its readability and comprehension.

lines 234-235: very bad sentence. rephrase it. 

- CONCLUSIONS:
no comments on this section. just be sure that the protocol you refer to is clearly evident from the remaining part of the paper.

- References are ok

Minor concerns:
- please avoid Nine(9), Three (3): either you use numbers or letters, choose one.
- please avoid massive citations [21-28], [24-28], and so on; please exploit references to precisely justify your statements. 
- figure captions must be improved and better detailed

Reviewer 2 Report

The presented work is somewhat innovative, and would be of interest to most readers. However, some technical issues need to be addressed. What were the detailed (objective) criteria for choosing the 3 apps? The overall patterns in the results were consistent across app evaluation. For example, participants with visual impairments had greater difficulty in completing the tasks with all 3 apps. This is known already. If the presented work is only presenting what is already generally known in app accessibility, then the innovation / contribution of the presented work is limited. It would be more interesting to know how the results compare to existing benchmarks (e.g. evaluation results using other apps). It is unclear how the qualitative results were obtained. Please specify clearly in Section 2 Materials and Methods. In Section 4, the discussed issues and recommendations are general. It would be better if the discussion can be more specifically based on the results of the presented work. The work only focused on evaluating weight management mHealth apps. The authors are encouraged to discuss the future directions, such as (i) whether disease management and medication compliance apps will be included, and (ii) existing and upcoming standards for mHealth apps (and how the authors' work can contribute in that)

Reviewer 3 Report

The authors have approached a very important topic nowadays, mHealth & accesibility. This is interesting for clinicians, institutions and patients. I have some suggestions in order to improve the manuscript.

Abstract: The abstract lacks information about the protocol per se. The reader doesn't know until later in the text what  the research was about.

Line 36: Please explain briefly what are the accesibility issues. Otherwise, the reader is forcer to look at the reference.

Material and Methods

Line 82-83: This information would be useful in the abstract.

Line 105: What were the adaptations?

Line 129: The n of the sample should be stated in the abstract. Being such a small number of participants, I strongly recommend including the denomination "pilot study" in the title.

Line 158: It is really determinant that most participants were white and non-Hispanic? Please explain further and consistently the relevance of this statement for the study or remove this information. 

Table 1: It is a really heterogeneus group, that will surely influence the results. Please include this in the limitations section.

Results: I find this section clear and well-constructed

Discussion: There is not any reference to other studies in the discussion. That makes it a no-discussion at all, but a guide for developers. All the discussion, as it is now, should be moved to the Future work section and the authors should rewrite a real discussion with other authors.

Author Response

Please see the attachment. Thank you for your time and feedback!

Round 2

Reviewer 2 Report

Reviewers' comments have been adequately addressed in this revision. 

Reviewer 3 Report

I think the authors did a extensive work in the manuscript and have adressed all my major concerns.